# RAM-PGK: Prediction of Lysine Phosphoglycerylation Based on Residue Adjacency Matrix

**DOI:** 10.3390/genes11121524

**Published:** 2020-12-20

**Authors:** Abel Avitesh Chandra, Alok Sharma, Abdollah Dehzangi, Tatushiko Tsunoda

**Affiliations:** 1School of Engineering & Physics, University of the South Pacific, Laucala Bay, Suva, Fiji; 2Laboratory for Medical Science Mathematics, RIKEN Center for Integrative Medical Sciences, Yokohama 230-0045, Japan; 3Institute for Integrated and Intelligent Systems, Griffith University, Brisbane, QLD 4111, Australia; 4Department of Computer Science, Rutgers University, Camden, NJ 08102, USA; i.dehzangi@rutgers.edu; 5Center for Computational and Integrative Biology, Rutgers University, Camden, NJ 08102, USA; 6Laboratory for Medical Science Mathematics, Department of Biological Sciences, Graduate School of Science, The University of Tokyo, Tokyo 113-0033, Japan; tsunoda@bs.s.u-tokyo.ac.jp; 7Department of Medical Science Mathematics, Medical Research Institute, Tokyo Medical and Dental University, Tokyo 113-8510, Japan

**Keywords:** post-translational modification, protein sequence, residue adjacency matrix, protein lysine modification database, amino acids, lysine, phosphoglycerylation, non-phosphoglycerylation, support vector machine, predictor

## Abstract

Background: Post-translational modification (PTM) is a biological process that is associated with the modification of proteome, which results in the alteration of normal cell biology and pathogenesis. There have been numerous PTM reports in recent years, out of which, lysine phosphoglycerylation has emerged as one of the recent developments. The traditional methods of identifying phosphoglycerylated residues, which are experimental procedures such as mass spectrometry, have shown to be time-consuming and cost-inefficient, despite the abundance of proteins being sequenced in this post-genomic era. Due to these drawbacks, computational techniques are being sought to establish an effective identification system of phosphoglycerylated lysine residues. The development of a predictor for phosphoglycerylation prediction is not a first, but it is necessary as the latest predictor falls short in adequately detecting phosphoglycerylated and non-phosphoglycerylated lysine residues. Results: In this work, we introduce a new predictor named RAM-PGK, which uses sequence-based information relating to amino acid residues to predict phosphoglycerylated and non-phosphoglycerylated sites. A benchmark dataset was employed for this purpose, which contained experimentally identified phosphoglycerylated and non-phosphoglycerylated lysine residues. From the dataset, we extracted the residue adjacency matrix pertaining to each lysine residue in the protein sequences and converted them into feature vectors, which is used to build the phosphoglycerylation predictor. Conclusion: RAM-PGK, which is based on sequential features and support vector machine classifiers, has shown a noteworthy improvement in terms of performance in comparison to some of the recent prediction methods. The performance metrics of the RAM-PGK predictor are: 0.5741 sensitivity, 0.6436 specificity, 0.0531 precision, 0.6414 accuracy, and 0.0824 Mathews correlation coefficient.

## 1. Introduction

Post-translational modification (PTM) is a biological process in which enzymatic change in proteins takes place, and this eventuates after the protein translation in the ribosome. Advancement of high-throughput proteomics from site-specific PTM and protein-altering enzymes has led to a stir of interest in the scientific community [1]. Lysine is one of the most heavily modified amino acids out of the 20 amino acids that form the genetic code [2,3]. Based on the reports [4], the lysine residues can quite easily go through covalent modifications. Some of the detectable covalent modifications are methyl [5], pupyl [6], succinyl [7], propionyl [8], crotonyl [9], acetyl [10], glycation [11] and glycosyl [12]. As a result of these modifications and changes to the regulatory enzymes, there have been a number of human diseases such as multiple sclerosis, neurodegenerative disorders, coronary heart diseases, rheumatic arthritis, coeliac disease, high blood pressure and essential hypertension [13,14,15,16].

Phosphoglycerylation is a type of non-enzymatic modification found in human cells and mouse liver tissues that can be classified as a newly discovered PTM [17,18]. It is highly connected to cardiovascular diseases like heart failure because of its link to the glycolytic process and glucose metabolism [19,20]. Phosphoglycerylation is a reversible process chemically known as 3-phosphoglyceryl-lysine (pgK), which results when primary glycolytic intermediate (1,3-BPG) reacts with the lysine residue [18]. This compound affects glycolytic enzymes and accumulates on cells with high exposure to glucose, and thereby creates a potential feedback process, causing buildup and redirection of glycolytic intermediates to alternate biosynthetic pathways. This new PTM, which is phosphoglycerylation, needs to be investigated more so that its identification and analysis becomes more apparent to recognize the selectivity mechanism and regulatory roles for the improvement of diagnosis and treatment procedures of those affected.

The identification of phosphoglycerylation and non-phosphoglycerylation sites using computational techniques is becoming popular [21,22,23,24,25,26,27,28,29,30,31,32,33,34,35]. This shift has been due to the high cost, time-consumption, and inefficiency involved while carrying out experimental methods in laboratories, such as mass spectrometry [36,37,38]. The computational methods have shown a lot of promise compared to the traditional way in the identification of the modified and unmodified sites.

In recent times, a number of computational techniques to identify phosphoglycerylation sites have been developed. One of the very first predictors is called Phogly-PseAAC, which employed pseudo amino acid composition for feature set and trained a k-nearest neighbors (knn) based predictor [39]. The next phosphoglycerylation predictor to be created was known as CKSAAP_Phoglysite, and it utilized the composition of k-spaced amino acid pairs (CKSAAP) for feature construction, and trained a fuzzy support vector machine (SVM) for prediction [17]. PhoglyPred is a method related to the CKSAAP_Phoglysite predictor, and it uses the protein sequence information obtained using various properties, such as the increment of k-mer diversity, the position-specific propensity of k-space dipeptide and the modified composition of k-space amino acid pairs with selected physicochemical attributes, to train an SVM classifier [40]. Later, three more methods for predicting protein phosphoglycerylation sites emerged. They are PhoglyStruct [35], EvolStruct-Phogly [41], and Bigram-PGK [42]. The most recent method, with a web server that allows users to query protein sequences for phosphoglycerylation sites, is called iPGK-PseAAC [43]. The iPGK-PseAAC considers four tiers of amino acid pairwise couplings as its features for segment size of seven upstream and seven downstream residues of the lysine. The resulting feature set is a 50-dimensional vector consisting of the frequency of occurrence of each pair, which was used to train an SVM classifier. Moreover, the work was conducted on a dataset which had 106 positive sites (phosphoglycerylation) and 1408 negative sites (non-phosphoglycerylation). Each of the predictors developed is claimed to have improved the prediction performance when compared to its predecessors, and hence offer further advantage for the scientific community in the study of protein phosphoglycerylation. The disadvantage, however, is that the earlier methods had smaller dataset available to build the predictor, while the more recent ones did not utilize the entire dataset for the validation of their respective methods.

Despite having a number of phosphoglycerylation site predictors, including the most recent webserver by the iPGK-PseAAC method, there still remains inadequate performance, and thus limits the computational technique in identifying the sites. To further improve the prediction, we introduce a new predictor called RAM-PGK that uses sequence-based information to classify phosphoglycerylation and non-phosphoglycerylation sites. In this work, 91 protein sequences are used that have experimentally confirmed sites, and these sequences were analyzed to obtain the residue adjacency matrix (RAM) corresponding to each lysine residue. Furthermore, the high class imbalance between positive and negative samples was a ratio of about 1:29, where the minority class was the phosphoglycerylated sites. To reduce the effect of this imbalance, some of the negative samples were filtered out in the training phase only, through the use of the k-nearest neighbor strategy [36,41,44,45]. The modified training sets through the k-nearest neighbor technique were then used to train the support vector machine (SVM) based RAM-PGK predictor, and it showed superior performance over the iPGK-PseAAC and CKSAAP_PhoglySite methods on sixfold cross-validation procedure. In the validation process, all of the samples of the dataset were used during the test phase to evaluate the performance of the predictor. The performance of RAM-PGK was 0.5741 on the sensitivity metric, 0.6436 on specificity, 0.0531 on precision, 0.6414 on accuracy, and 0.0824 on the Mathews correlation coefficient.

## 2. Materials and Methods

### 2.1. Protein Dataset

The dataset used in this work is a benchmark dataset obtained from CPLM repository (Compendium of Protein Lysine Modifications, http://cplm.biocuckoo.org) now newly known as Protein Lysine Modification Database (PLMD). Phosphoglycerylation is among the many other lysine modifications that have been experimentally identified in protein sequences and uploaded on PLMD. To use the phosphoglycerylation dataset for this work, the protein sequences which had 40% or higher sequential similarity were removed using the Cd-hit tool. The resulting dataset had 91 sequences, from which, a total of 3360 lysine residues were obtained. These lysine residues were then investigated and found to contain 111 phosphoglycerylation sites and 3249 non-phosphoglycerylation sites. The datasets used and analyzed during the current study are publicly available online at https://github.com/abelavit/RAM-PGK or www.alok-ai-lab.com.

### 2.2. Residue Adjacency Matrix

RAM is constructed from n-nearest amino acid distances sequentially from the lysine residues. RAMseq (Residue Adjacency Matrix from sequence data) is an alias of RAM and can simply be calculated from raw protein sequence without the need for any other information like 3-D coordinates or the secondary structure [46]. The matrix is of 20 by n dimension, which is a result of taking the distance of the target lysine residue to the first, second, third, fourth, and n-th closest amino acid residue of each type in the sequence. The value of n in this work is taken as six, since it is anticipated to work well on different types of datasets [46]. The distance in RAM is obtained by taking the index of the amino acid and subtracting it with the index of the lysine residue, up to the n-th amino acid index of the same type. The absolute values are then taken to make the distances positive despite the amino acid in question existing before or after the lysine residue.
(1)$$RAM_{i,j}=\left|AA_{j}^{i}-K\right|$$

In Equation (1) above, the value of RAM is the residue adjacency matrix, *K* is the index of lysine in question, and *AA* corresponds to the amino acid index. Moreover, *i* is the amino acid type and *j* is the *j*-th nearest amino acid of the same type. This formulation is depicted in Figure 1. A *j* value of 1 (1st column) and *i* value of 1 (1st row) indicates the nearest amino acid of the type Alanine (A). If *j* = 3 (3rd column) and *i* = 19 (19th row), it refers to the third nearest amino acid of the type Tyrosine (Y). In the case when a particular type of amino acid is not enough to fill the matrix, the average value of the available amino acid distances of the same type is used. Furthermore, if an amino is not found altogether, the mean of the entire entries of the matrix is used to fill in its row. These two cases are portrayed in Table 1. In the table, n is taken as 3. The mean of column *K* is (0 + 2)/2 = 1 and *Y* is (1 + 1 + 3 + 0 + 2 + 2)/6 = 1.5. Here, Tyrosine (Y) is missing completely in the sequence. Therefore, the average of the entire matrix is calculated, while the Lysine (K) is missing for the third nearest residue hence the average is calculated from the lysine residues which are present.

### 2.3. Support Vector Machine

A support vector machine is a supervised learning algorithm in the area of machine learning. The algorithm is well known for its application in classification and regression problems. SVM algorithm is a discriminative classifier that generates a hyperplane which best separates the two classes in the training dataset. Then it uses this hyperplane to categorize every new data point (test data) presented to it. For every data point of a two-class problem, it represents a point in an n-dimensional space, where n represents the number of features, which belongs to either of the classes that are not always linearly separable. In these cases, a nonlinear kernel is used for the classification purpose where it projects the nonlinear input space to a higher dimensional space that allows them to be linearly separated. In this work, RAM-PGK was developed based on the LibSVM package for the Matlab software in order to identify phosphoglycerylation and non-phosphoglycerylation sites. The type of SVM employed is the C-SVC type and the kernel used is linear.

### 2.4. Feature Selection Scheme

In our work, we carried out a successive feature selection (SFS) technique to select amino acids from a residue adjacency matrix which contribute favorably towards the identification of phosphoglycerylation and non-phosphoglycerylation sites. The type of SFS scheme we used is called backward elimination [47]. In this method, a row of a residue adjacency matrix which corresponds to a particular type of amino acid (refer to Figure 1) is permanently removed (eliminated) at each successive level from the feature matrix. At each level, the removal of an amino acid row that results in the highest sensitivity metric on 6-fold cross-validation using the SVM classifier is permanently removed from the feature matrix and progressed to the next subsequent level. The removal of a row resulting in highest sensitivity measure for the level denotes that its absence is more advantageous. Moreover, the elimination of a row at each level causes the feature vector, obtained from the feature matrix, to reduce by a size of 6 as the network is progressed.

Figure 2 shows the performance of the features on the sensitivity metric as the backward elimination feature selection technique is carried out. As can be seen, the highest performance (0.5741) is achieved on the sixth level, when the adjacency rows of Glutamic Acid (E), Aspartic Acid (D), Leucine (L), Valine (V), Phenylalanine (F), and Threonine (T) are eliminated from the feature matrix. Further elimination of amino acid adjacency rows does not surpass this measure. Thus, the resulting feature matrix used for building the RAM-PGK classifier by eliminating the six amino acid adjacency rows is of the size 14 × 6, which when converted to a feature vector is 84-dimensional.

## 3. Results and Discussion

### 3.1. Dataset Balancing

The dataset of this work was imbalanced when obtained from PLMD. There were only 111 positive samples (phosphoglycerylation sites) compared to 3249 negative samples (non-phosphoglycerylation sites) and this resulted in an imbalance ratio of 1:29. This high imbalance ratio is bound to bias the classifier towards the majority class (non-phosphoglycerylation); hence it is important to resolve the imbalance issue to build an effective classifier. The k-nearest neighbors (knn) cleaning treatment was utilized, which removed samples from the negative dataset when the samples were one of the N neighbors of any positive sample. It is important to note here that this cleaning treatment was employed for the model training purpose only, and the evaluation of the model’s performance took place on the entire dataset. The knn cleaning treatment was initiated with an N value of 29, which was the imbalance ratio. It was, however, found that the imbalance remained high, with an N value of 29. This threshold was increased further until the imbalance ratio was close to 1:2, so, for every positive sample, there were about two negative samples. The N value of 101 reduced the imbalance ratio to about 1:2.

### 3.2. Statistical Measures

Statistical measures help evaluate the performance of classifiers. To deduce the performance of the predictor proposed in this work, we used five statistical measures, namely sensitivity, specificity, precision, accuracy, and Mathews correlation coefficient (MCC) [17,36,39,45,48,49,50,51,52].

The sensitivity metric determines the ability of a predictor to identify phosphoglycerylated lysine sites correctly. The value of this measure ranges between 0 and 1, where the higher the value, the better the predictor at classifying positive sites. The second metric is specificity, and it determines the ability of the predictor to correctly identify non-phosphoglycerylated lysine sites. The range of values of this metric is also between 0 and 1, and the higher the value, the better the predictor at predicting negative sites. The next two metrics are precision and accuracy, respectively, and like sensitivity and specificity, they take on values ranging from 0 to 1. Out of all the phosphoglycerylated sites classified by a predictor, the extent of the correctly classified phosphoglycerylation sites is determined by the precision metric. Furthermore, the ability of a predictor to distinguish between phosphoglycerylated and non-phosphoglycerylated sites is evaluated by the accuracy metric. The final metric is Mathews correlation coefficient, and it determines the predictor’s quality. *MCC* takes on values from −1 to +1, where −1 corresponds to an entirely negative correlation, while a +1 corresponds to an absolutely positive correlation. The five statistical measures can be written in terms of equations as:(2)$$Sensitivity=\;\frac{TP}{TP+FN}$$
(3)$$Specificity=\;\frac{TN}{TN+FP}$$
(4)$$Precision=\;\frac{TP}{TP+FP}$$
(5)$$Accuracy=\;\frac{TN+\;TP}{FN+FP+\;TN+TP}$$
(6)$$MCC=\;\frac{\left(TN\times TP\right)-\left(FN\times FP\right)}{\sqrt{\left(TP+FP\right)\left(TP+FN\right)\left(TN+FP\right)\left(TN+FN\right)\;}}$$

From the Equations (2)–(6), *TP*, *FN*, *TN*, and *FP* represent true positives, false negatives, true negatives, and false positives, respectively. True positives are samples which have been correctly classified as phosphoglycerylation sites. False negatives are phosphoglycerylation sites that have been classified as non-phosphoglycerylation sites by the predictor. True negatives are the non-phosphoglycerylation samples correctly classified by the predictor. Finally, false positives are non-phosphoglycerylation samples that are classified as phosphoglycerylation samples. While comparing predictors, it is desired for the best predictor to have high scores in all the statistical measures. In addition, the proposed predictor should at least have high scores in the majority of the statistical measures when compared to the existing predictors.

### 3.3. Test Scheme

The assessment of our predictor’s performance was carried out via the sixfold cross-validation scheme. The statistical measures discussed in the previous section were calculated for each fold and were then finally averaged to determine the predictor’s overall performance. The validation scheme of this work is generally known as the n-fold cross-validation scheme. Apart from this, there are two other well-known schemes to determine the effectiveness of a predictor, which are called the independent dataset test and the jackknife test [53,54]. From the three validation schemes, the jackknife scheme is said to be the least arbitrary, whereby it yields distinct results on the dataset [55]. However, we employed the n-fold cross-validation scheme primarily to avoid the high computational time. The sixfold cross-validation scheme was carried out using the steps highlighted below:Step 1: Divide the dataset into six similar parts.Step 2: Combine the five parts and apply the cleaning treatment to balance the positive and negative classes. Train the predictor using this balanced dataset and test it with the part left out.Step 3: Set the predictor parameters with the train set.Step 4: Acquire the statistical measures on the test set.Step 5: Repeat steps 2 to 4 for the remainder of the folds.

### 3.4. Comparison of RAM-PGK with iPGK-PseAAC Predictor

The iPGK-PseAAC predictor [43] is the most recent and is claimed to be the most effective method in distinguishing phosphoglycerylated and non-phosphoglycerylated sites. iPGK-PseAAC predictor has a web server that allows protein sequences to be uploaded in FASTA format to identify whether the lysine residues are phosphoglycerylated or not. The CKSAAP_PhoglySite is another predictor which was introduced before the iPGK-PseAAC predictor and it allows users to identify phosphoglycerylation sites via the use of Matlab software package. It is important to note that the protein sequences used for performance evaluation in this work may have been used to train the iPGK-PseAAC and CKSAAP_PhoglySite predictors, and as a result, the output of their method to compute the performance when comparing with RAM-PGK can be biased in their favor. To avoid this, we replicated the iPGK-PseAAC and CKSAAP_PhoglySite methods by constructing their features, as outlined in [43] and [1] respectively, and used the same train and test samples, as well as the same classifier as our method to obtain the performances. The performance was calculated using the sixfold cross-validation scheme.

Table 2 shows the comparison of the iPGK-PseAAC and CKSAAP_PhoglySite methods with RAM-PGK. In the same manner, we also compared the performance of RAM-PGK with our previous work (Bigram-PGK) that utilized PSSM + bigram features to predict phosphoglycerylation. From the table, it can be seen that RAM-PGK outperforms both the iPGK-PseAAC and CKSAAP_PhoglySite methods, as well as the Bigram-PGK method on the three metrics: sensitivity, precision, and MCC. The sensitivity measure is increased by 29.4%, precision by 19.4%, and MCC by 64.6%. It is to be noted that these improvements are referenced from the Bigram-PGK method since iPGK-PseAAC and CKSAAP_PhoglySite methods did not perform well on those metrics. Moreover, the MCC value of the iPGK-PseAAC method is less than zero. These notable improvements in performance suggest that RAM-PGK is a lot more useful than the three methods, and especially over the iPGK-PseAAC and CKSAAP_PhoglySite methods in dealing with the phosphoglycerylation classification problem.

In addition to Table 2, we compared the time needed to carry out feature construction and sixfold cross-validation of each of the methods on the Matlab software. The result can be obtained from the Appendix A. The iPGK-PseAAC method took the least time of 15.6 s, the RAM-PGK took 49.01 s, CKSAAP_PhoglySite took 486.35 s, while the Bigram-PGK method took 5096.22 s. The reason for Bigram-PGK having a large time is due to the feature construction using the PSI-BLAST tool [56].

To further demonstrate the superiority of the RAM-PGK method, we calculated the area under the receiver operating characteristic curve (AUC) for each of the methods of Table 2. The Receiver Operating Characteristic (ROC) curves are displayed by Figure 3.

The result of Table 2 implies that the RAM-PGK has a better capacity to identify phosphoglycerylated and non-phosphoglycerylated lysine residues correctly. The use of a residue adjacency matrix of amino acid residues in proteins can be credited for this improvement. The RAM of neighboring residues of the lysine residue and the transformation of this matrix into feature vector has proven to produce necessary characteristics for differentiating the modified lysine residues. Moreover, the performance enhancement can also be credited to the SVM algorithm as it has been effective in handling the data comprising the sequential information.

## 4. Conclusions

This work carried out has proposed a new predictor called RAM-PGK, which efficiently uses a residue adjacency matrix feature set for phosphoglycerylation prediction. In the approach of phosphoglycerylation prediction, using the sequential information concealed in RAM was shown to have good potential that can be taken into consideration for identifying the PTM. To balance the dataset, k-nearest neighbors cleaning treatment was used in removing the negative samples during the training stage. A reasonably balanced dataset in the training phase and the entire samples for testing, alongside a support vector machine algorithm, provided the RAM-PGK with the thoroughness to outperform some of the latest phosphoglycerylation prediction methods in the literature.

## Figures and Tables

**Figure 1 genes-11-01524-f001:**
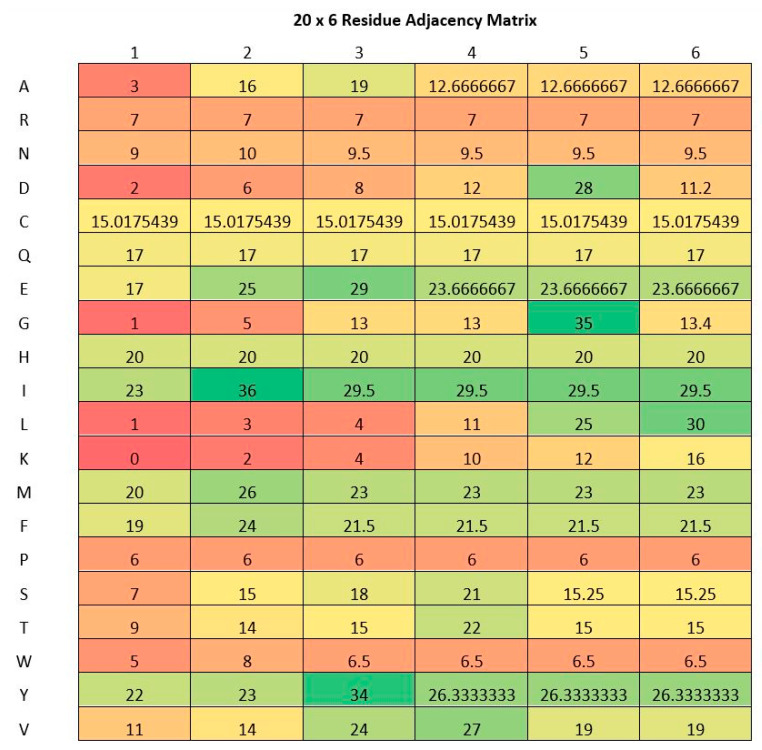
A residue adjacency matrix computed from lysine residue indexed at 27 of Acyl-CoA-binding protein (UniProt Accession: D3Z563).

**Figure 2 genes-11-01524-f002:**
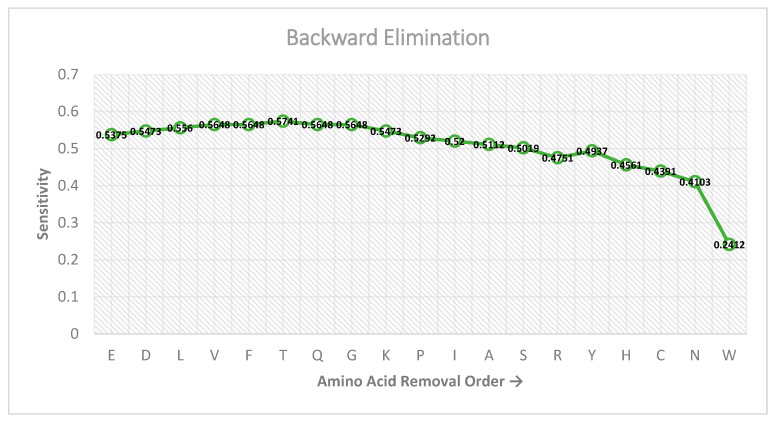
Result of the backward elimination feature selection scheme at each successive level.

**Figure 3 genes-11-01524-f003:**
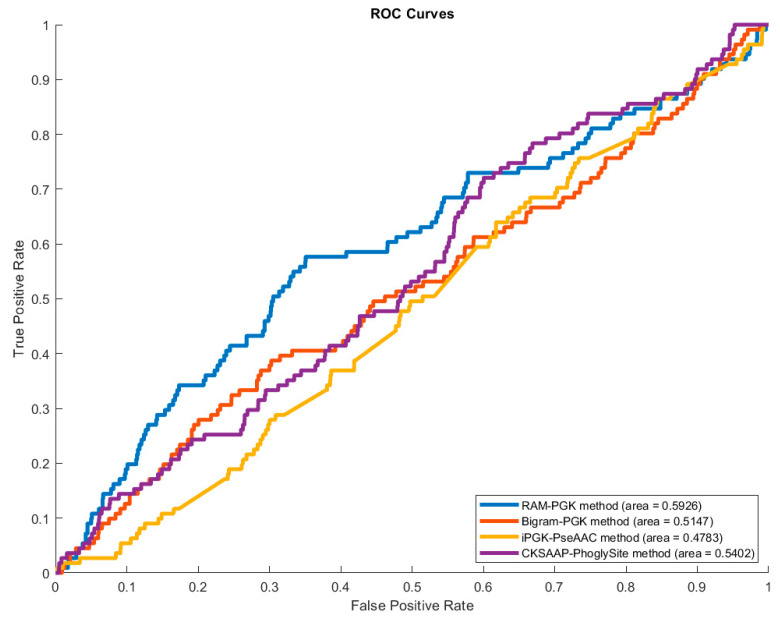
Plot of area under Receiver Operating Characteristic (ROC) curve of the four methods.

**Table 1 genes-11-01524-t001:** Illustration of residue adjacency matrix construction using a dummy protein sequence ‘MAKAKAA’ with n equal to 3. The analysis is shown for the first lysine residue. The top side of the table (in yellow) shows a matrix without consideration for the missing amino acid residues. The bottom side of the table (in green) shows a complete residue adjacency matrix with consideration for the missing amino acid residues.

	1	2	3
A	1	1	3
K	0	2	N/A
M	2	N/A	N/A
Y	N/A	N/A	N/A
A	1	1	3
K	0	2	1
M	2	2	2
Y	1.5	1.5	1.5

**Table 2 genes-11-01524-t002:** Comparison of the iPGK-PseAAC, CKSAAP_PhoglySite and Bigram-PGK methods with the RAM-PGK predictor using 6-fold cross-validation scheme. Highest values of the metrics are highlighted in bold. MCC: Mathews correlation coefficient.

Predictor	Sensitivity	Specificity	Precision	Accuracy	MCC
CKSAAP_PhoglySite [17]	0.3494	0.6722	0.0358	0.6616	0.0090
iPGK-PseAAC [43]	0.0185	0.9791	0.0064	0.9473	−0.5048
Bigram-PGK [42]	0.4055	0.6639	0.0428	0.6554	0.0292
RAM-PGK (No Feature Selection)	0.5380	0.6328	0.0472	0.6298	0.0631
RAM-PGK	0.5741	0.6436	0.0531	0.6414	0.0824

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
