# Peer review of "RAM-PGK: Prediction of Lysine Phosphoglycerylation Based on Residue Adjacency Matrix"

_genes, 2020, doi:10.3390/genes11121524_

Round 1
Reviewer 1 Report
In this article, the authors proposed a tool named RAM-PGK, which based on SVM classifier with linear kernel, for identification of phosphoglycerylated lysine residues by extracting the residue adjacency matrix pertaining to each lysine sites in the protein sequences. Although the results of test dataset show relatively satisfying performance, there are some questions should be conducted before further consideration. Detailed comments are as follows:
- Introduction: there are numerous predictors of phosphoglycerylation residues have been reported in the past few years. The "Introduction" has few introductions for these aspects not to highlight this innovation of this paper. The authors should briefly summary the advantage and disadvantage of these predictors so as to deliver the innovation of this method.
- Regarding the protein dataset, there lacks of some reported phosphoglycerylation sites. Furthermore, the tool for removing the sequential similarity with 40% or higher should be provided so that other researchers can repeat the results.
- The robustness of RAM-PGK based on six-fold cross-validation displayed by ROC curves so that the performance visually displayed. Furthermore, there are no more details for dimensions of the optimal model in this paper, more information on the selection of different feature dimension should be provided.
- The knn cleaning treatment was employed to reduce the imbalance, the similarity of negatives clustered together whether it can enhance the performance comparing with downsampling method?
- How to select the values of N, and what is the reason for choosing N as 6 and what are the considerations? The details of the selection of N value in knn cleaning treatment are not found.
- Although the authors believed that the performance of RAM-PGK outperformed the existing state-of-the-art methods via comparing with iPGK-PseAAC, there is no an online web services or applications for researchers to examine the prediction results of RAM-PGK. We encourage the authors to develop a user-friendly online tool for public.
Author Response
Point-by-point response
1. Introduction: there are numerous predictors of phosphoglycerylation residues have been reported in the past few years. The "Introduction" has few introductions for these aspects not to highlight this innovation of this paper. The authors should briefly summary the advantage and disadvantage of these predictors so as to deliver the innovation of this method.
Thanks for this comment. We have now highlighted in the Introduction section other predictors which have been proposed recently. The advantage offered by every new predictor proposed is the improvement of prediction performance when compared to its predecessor. The disadvantage however is that the earlier methods had smaller dataset available to build the predictor, while the more recent ones did not utilize the entire dataset for the validation of their respective methods. Even though the latest predictor (iPGK-PseAAC) is claimed to be the most effective predictor, it still does not have high performance and so the phosphoglycerylation prediction can be improved further by introducing new methods and as a result, we are introducing a new predictor called RAM-PGK.
2. Regarding the protein dataset, there lacks of some reported phosphoglycerylation sites. Furthermore, the tool for removing the sequential similarity with 40% or higher should be provided so that other researchers can repeat the results.
Thanks for pointing this out. We have now added the name of the tool which is Cd-hit.
3. The robustness of RAM-PGK based on six-fold cross-validation displayed by ROC curves so that the performance visually displayed. Furthermore, there are no more details for dimensions of the optimal model in this paper, more information on the selection of different feature dimension should be provided.
I appreciate this comment. The AUC values of RAM-PGK and the methods it has been compared to are now calculated and displayed using the ROC curves (Figure 3). We also have incorporated the feature selection scheme in our work as depicted by Figure 2. The backward elimination feature selection scheme has been employed and the resulting dimension of the feature matrix used for building the RAM-PGK classifier is 14 × 6 (feature vector dimension of 84), from the originally used feature matrix of size 20 × 6 (feature vector dimension of 120).
4. The knn cleaning treatment was employed to reduce the imbalance, the similarity of negatives clustered together whether it can enhance the performance comparing with downsampling method?
Thanks for the question. The knn cleaning treatment is a form of downsampling method.
5. How to select the values of N, and what is the reason for choosing N as 6 and what are the considerations? The details of the selection of N value in knn cleaning treatment are not found.
Thanks for the questions. In our work, label ‘n’ is dealing with the feature matrix, while ‘N’ is the value for knn cleaning treatment. The value of n used for constructing the feature matrix is 6. The value of 6 is used since it was suggested in the original paper which introduced the residue adjacency matrix based feature engineering. Moreover, the selection of N for the knn cleaning treatment is explained under the subsection “Dataset balancing”.
6. Although the authors believed that the performance of RAM-PGK outperformed the existing state-of-the-art methods via comparing with iPGK-PseAAC, there is no an online web services or applications for researchers to examine the prediction results of RAM-PGK. We encourage the authors to develop a user-friendly online tool for public.
Thanks for the comment. We have uploaded our data and codes on github and they can be obtained from https://github.com/abelavit/RAM-PGK. Using this repository, the researchers can examine and replicate the results. We have also included a ‘Readme’ file that will assist the researchers on how to use the codes.
Reviewer 2 Report
The manuscript describes RAM-PGK tool, which allows to predict lysine phosphoglycerylated and non-phosphoglycerylated sites. Although results look interesting, authors should make comparison with more available tools and clearly state what is new and innovative (novelty) in their approach.
Major comments:
Explicit comparison of obtained results, with other tools should be provided in Table 2, e.g. other method of the authors: „Bigram-PGK: phosphoglycerylation prediction using the technique of bigram probabilities of position specific scoring matrix” and methods of other groups, e.g. „iDPGK: characterization and identification of lysine phosphoglycerylation sites based on sequence-based features”. Additionaly, comparison of time needed to perform prediction by different tools should be provided.
Minor comments:
“The traditional method of 21 identifying phosphoglycerylated residues, which are experimental procedures,” – what experimental procedures are?
“This new PTM needs to be investigated more” – type of the PTM should be given explicitly.
Author Response
Point-by-point response
Major comments:
Explicit comparison of obtained results, with other tools should be provided in Table 2, e.g. other method of the authors: „Bigram-PGK: phosphoglycerylation prediction using the technique of bigram probabilities of position specific scoring matrix” and methods of other groups, e.g. „iDPGK: characterization and identification of lysine phosphoglycerylation sites based on sequence-based features”. Additionaly, comparison of time needed to perform prediction by different tools should be provided.
Thank you for this comment.
- Table 2 has now been modified to include the previous method of the authors, namely the Bigram-PGK method.
- The method for other group “iDPGK: characterization and identification of lysine phosphoglycerylation sites based on sequence-based features” has not been considered since the work is still under review and is not a published work yet.
- We have now carried out comparison of the time needed to carry out feature construction and 6-fold cross-validation of each of the methods of Table 2. The comparison was carried out on a machine with 24 CPUs (Intel(R) Xeon(R) CPU E5 – 2650 v4 @ 2.20GHz) and 125.8 GB memory. PSI-BLAST tool was run using 16 CPUs while the other programs was run on single CPU. This can be found in Supplementary Material and is mentioned in the Results and Discussion section.
Minor comments:
“The traditional method of identifying phosphoglycerylated residues, which are experimental procedures,” – what experimental procedures are?
Thanks for pointing this out. An example of the experimental procedure have now been added.
“This new PTM needs to be investigated more” – type of the PTM should be given explicitly.
Thanks for the comment. The PTM has been explicitly stated.
Round 2
Reviewer 2 Report
Authors fixed some of the issues raised by the Reviewers, however, the main issue (novelty) remained not sufficiently explained. As the methods used to obtain a new tool to predict phosphoglycerylated lysines are not novel, authors have to show without doubt that obtained tool is better than all other methods available. Authors published several similar works in the past and results from these and other methods have to be extensively compared to the new method described in this work. In paper "PhoglyStruct: Prediction of phosphoglycerylated lysine residues using structural properties of amino acids" authors compared performance with following methods: CKSAAP_PhoglySite, Phogly-PseAAC, and PhoglyStruct and results from these methods have to be presented in Table 2 of the manuscript. Moreover, iDPGK method was already published as preprint and it is available for users before authors submitted original version of this manuscript, so there are no reasons why it should not be included in evaluation as well.
To allow easier reproduction of the results, all files used to train and test the method should be placed at github.
Author Response
Point-by-point response
Authors fixed some of the issues raised by the Reviewers, however, the main issue (novelty) remained not sufficiently explained. As the methods used to obtain a new tool to predict phosphoglycerylated lysines are not novel, authors have to show without doubt that obtained tool is better than all other methods available. Authors published several similar works in the past and results from these and other methods have to be extensively compared to the new method described in this work. In paper "PhoglyStruct: Prediction of phosphoglycerylated lysine residues using structural properties of amino acids" authors compared performance with following methods: CKSAAP_PhoglySite, Phogly-PseAAC, and PhoglyStruct and results from these methods have to be presented in Table 2 of the manuscript. Moreover, iDPGK method was already published as preprint and it is available for users before authors submitted original version of this manuscript, so there are no reasons why it should not be included in evaluation as well.
To allow easier reproduction of the results, all files used to train and test the method should be placed at github.
Thank you for the comment.
For a fair comparison, we are comparing with other methods by implementing their methods and then use the same training and test samples, as well as the classifier as our method to obtain the performances. Moreover, some of the works have not provided the feature set and algorithm for others to replicate and check their results, which is a setback, and replicating their work takes a long time and even impossible in some cases. Result from their webserver do not give a true indication of the performance of those methods as it is difficult to find out which training samples they have used to train their model. This gives bias towards their models. In the short timeframe given to revise this paper, we have managed to carry out comparison with one other method, which is the CKSAAP_PhoglySite. Furthermore, we do have our training and test files with feature set and classifier on the GitHub and provided the link in the paper.